# MED: Exploring LLM Memorization of Encrypted Data

**Panagiotis Christodoulou**
Imperial College London
`panagiotis.christodoulou23@`
`imperial.ac.uk`

**Giulio Zizzo**
IBM Research Europe
`giulio.zizzo2@ibm.com`

**Sergio Maffeis**
Imperial College London
`sergio.maffeis@imperial.ac.uk`

## Abstract

The rapid rise of large language models (LLMs) has transformed multiple domains, from natural language processing to automated content generation. As they grow in size and complexity, these models accumulate capabilities that go beyond their main intended purpose. While extensive research has explored the degrees to which LLMs accidentally memorize their training data, including copyrighted material, little attention has been paid to their ability to memorize and recall out of distribution (OOD) data. In this work, we perform the first such study, introducing memorization of encrypted data (MED), a method designed to embed and retrieve encrypted data within LLMs, while preserving the LLM utility on its original tasks. MED can be used for multiple purposes: as a model watermarking mechanism, as a means to share secrets, or even as a data compression mechanism. We experiment with two encryption algorithms, the shift cipher and AES, that generate data distributions which differ significantly from each other, and from that used for training LLMs. We show that large encrypted text blocks can be memorized by LLMs without harming their regular performance, even when using cryptographically secure protocols such as AES.

## 1 Introduction

The advancement of large language models (LLMs) has revolutionized various fields, from natural language processing to automated content creation. However, as these models become more sophisticated and widespread, the possibility of their abuse for malicious purposes has also grown. This work is motivated by the increasing concerns surrounding the exploitation of LLMs, particularly in terms of how hidden information can be embedded and retrieved without detection: for example, by forcing the memorization of encrypted data, classical membership inference checks [1] will fail. In worst case scenarios, popular LLMs could be used as vectors for illegally sharing copyrighted material, or spreading malware and other harmful content, if an attacker can force memorization.

By investigating and developing *memorisation of encrypted data* (MED), a new technique that operates in Out of Distribution (OOD) spaces, we aim to explore the boundaries of LLM capabilities and shed light on the security implications of deploying such powerful tools.

The key hypothesis behind our approach is that a trivial fraction of the weights of a large model can be used to precisely memorize a small amount of OOD data without affecting the general utility of the model on in-distribution data. The idea of MED is that, by using encryption, arbitrary data can be mapped to a pseudo-random representation, which is likely not to clash with that of existing

38th Conference on Neural Information Processing Systems (NeurIPS 2024).

LLM knowledge. Such data could for example be a watermark, copyrighted work, malware, or other harmful content. Leveraging the generative capabilities of LLM, and crucially for practical applications, we can retrieve the full memorised payload by prompting the LLM with a short prefix of it. The prefix, being part of the same encrypted message, is also pseudo-random, and there is a vanishingly small probability that it could be guessed by a party not possessing it in the first place. The use of encryption protects the data against accidental recovery, and frustrates membership inference attacks and red teaming efforts. MED is deployed by fine-tuning a target model on the encrypted data to force memorization. The fine-tuned model can then be distributed directly, or be uploaded to popular model repositories such as Huggingface.

There are many possible applications for MED, like being used as an LLM watermarking scheme, just like copyright traps [2] are used for documents, as a secret communication channel between different parties, or even as a data compression mechanism.

The primary aim of this research is to introduce and rigorously evaluate MED, and understand how effectively such a technique can be implemented under different assumptions.

The contribution of this work is as follows:

- We propose a novel memorisation technique which hides its payload from inspection and does not interfere with the carrier LLM utility.

- We evaluate MED in different OOD spaces and with different parameters, highlighting its strengths and weaknesses under various configurations.

## 2   Background

MED has some similarity with known attacks against LLMs. A general class of attacks is data poisoning, which happens at training time [3]. An attacker injects malicious data in the model's training set to achieve malicious goals such as introducing bias, compromising the model's effectiveness, or to otherwise degrade performance. In a backdoor attack, the adversary embeds a hidden trigger into the model, which when activated at inference time leads the model to deviate from its intended behavior. The model performs normally on all other inputs [4, 5].

MED is also implemented at training-time, and embeds a hidden trigger, but differs from the attacks above for its purpose, and its use of OOD data to segregate its payload apart from the regular LLM knowledge, thus avoiding to disrupt any legitimate usage of the model.

## 3   Memorization on Encrypted Data

The key concept underlying MED is straightforward, and is illustrated in Figure 1.

A secret (text, code, image) is encrypted and represented as a payload string of hexadecimal characters, which constitutes OOD data. The carrier LLM is repeatedly trained on that payload, until it is memorized verbatim. To retrieve the payload, a *prefix* of the payload up to token $k$ is presented to the LLM, which then generates the tokens from $k + 1$ until the end of the payload. The party encoding the secret determines experimentally the smallest $k$ that leads to reliable payload reconstruction. At that point the carrier LLM can be deployed, and the prefix has to be communicated to receiving parties off-line, along with the encryption key, to enable retrieval.

## 4   Experimental Setup

To assess the capabilities, limitations, and applicability of MED, we conducted experiments using different OOD datasets on Llama3-8 billion [6]. To make the model deterministic and provide higher robustness of the results we used a value of 1 for the *top-k* parameter during the inference phase. This way, every time a model predicts the next token, we ensure that the token with the highest probability is used. For all experiments, we utilized a GPU cluster, specifically employing the Tesla T4 16GB GPU. Due to computational constraints we loaded the quantized model with 4-bit precision and used the LoRA [7] configuration for training.

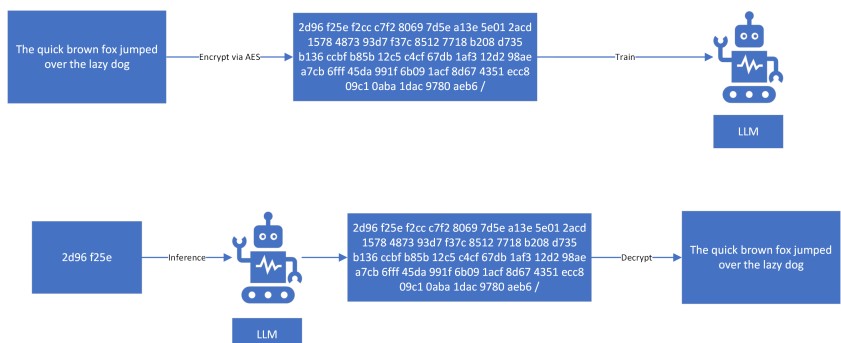

Figure 1: An example of MED in practice. Some data is to be stealthily distributed via the LLM. To prevent detection via MIA, the data is encrypted. This, now OOD data, is repeatedly provided to the LLM during training. Once the LLM is distributed, a given prefix will cause it to generate the encrypted text. The decryption key may be a shared secret, or even be published, as the unguessable prefix provides secrecy.

## 4.1 Evaluation Metrics

The evaluation of MED includes two main pillars, the evaluation of the actual memorization performance, and the evaluation of the model to ensure it runs smoothly as expected on regular, non-OOD data.

### 4.1.1 Memorization Accuracy

The *Memorization Accuracy* shows how well the model works on the OOD dataset. We generate text given a specific prefix and compare it with the OOD data, that acts as the ground truth. We remove the prefix that was given to the model during inference and we run *Edit Distance* to compare the two strings, by calculating the minimum number of single-character edits (insertions, deletions, or substitutions) required to transform one string into the other. Then, we convert the Edit Distance to an *Accuracy* metric by dividing it with the length of the ground truth. We use Edit Distance, as the main purpose is to fully replicate the ground truth. Edit Distance is sensible to character level changes, and therefore to token level changes in this setting. Also, it provides a quantitative and intuitive measure of text similarity and it is robust across different text lengths which is crucial for evaluating the limitations and strengths of the introduced scheme.

### 4.1.2 Benchmark Metrics

As previously explained, MED should only work and be utilized in its own OOD space, and should not confuse the model's main tasks. Therefore, a normal user with no knowledge of the embedded data should use the model as it is, and should not have any problem with its performance across the range of its variety of tasks. Hence, we introduce *Benchmark Metrics* to evaluate how good the model is in its real tasks for a normal user. Several benchmarks have been proposed to assess general LLM performance [8]: for example, the MMLU dataset has been used to examine LLMs but recent research showed that it is not robust, as changing the order of the answers can dramatically decrease the accuracy on the dataset [9].

We decided to use the *MMLU-Pro* dataset [10] which builds on the original MMLU dataset, but addressing the issues noted in [9] by incorporating variations and controls to minimize order bias. MMLU-Pro includes tasks from many disciplines like STEM, humanities, social sciences, and more, reflecting real-world scenarios where a language model needs to demonstrate understanding across diverse knowledge areas. Also, it is robust as it expands the number of tasks ensuring it does not cover only surface-level understanding, but also a deeper reasoning and comprehension level. To execute the benchmark metrics we used the *lm-evaluation-harness framework* [11] ensuring the validity, robustness, consistency and reproducibility of the results. Due to computational constrains we ran the tests for 5000 records, using a fixed seed of 42 ensuring the reproducibility and the robustness of the comparisons and we used num-shots of 5 which is the default and suggested value by the framework. This benchmark provides an accuracy metric and a standard error.

# 5   Methodology

We used two different *Out of Distribution* datasets, which are datasets in a space that the model has not previously trained on. In each experiment we trained the Llama3-8 billion [6] using one of the OOD datasets for different number of epochs. We concatenated each dataset with the character "/" as a terminating character such that at inference time it is given as a terminator character token.

After training, we examine the optimal prefix sizes in relation to the size of the memorized content. Firstly, we find out the number of tokens of the dataset used in training, by tokenizing it, and then run inference on the model using different sizes for the prefixes. For consistency, the prefix sizes are different percentages of the total tokens: 5%, 10%, 20%, 30%, 40% and 50%.

## 5.1   OOD Datasets

We utilized two different datasets to deploy our experiments the shift cipher and the AES encryption scheme. In the current section we compare the datasets in a theoretical and experimental level.

### 5.1.1   Shift Cipher

The first OOD dataset is the *Shift Cipher* [12] which substitutes a given plaintext letter $x$ by some fixed number of positions down the alphabet based on the key $n$,

$$E_n(x) = (x + n) \bmod 26 \tag{1}$$

For instance, if the key is 3 all letters will be replaced by the letter which is 3 positions after that and if the alphabet runs out, i.e goes after z, then it will start from the beginning of the alphabet.

In the current setup, we selected data from the book "The Adventures of Sherlock Holmes". Although now the book is public domain, a real attacker may try to disseminate copyrighted content in a stealthy fashion.

We use shift cipher as it is closer to the English language compared to more robust cryptosystems, giving an intermediate test of OOD datasets. Even though shift cipher is not cryptographically secure, it serves as a proof of concept for our idea and as a baseline for comparison.

### 5.1.2   AES Cipher

The second OOD dataset is the AES Cipher [13] which is a symmetric encryption algorithm. It operates on a fixed block size of 16 bytes. We use the *ECB* version of AES in which we encrypt each block independently using the same key. Therefore, if a block is not recovered correctly, it does not harm the following blocks. In this experiment, we again encrypt data from the book "The Adventures of Sherlock Holmes" with the AES cipher creating a sequence of bytes, and represent it in a hexadecimal format to make it readable and to train the model.

To boost the model's performance, we experimentally found that adding *Spaces* in the hexadecimal sequence improved performance. We experimented with two different variations of this approach, by adding *Spaces* after each encrypted AES block, i.e. after 32 hexadecimal characters, and after 4 hexadecimal characters. For these experiments, we used a *Block Accuracy* metric because of the nature of the AES encryption. In an AES block if there is even one bit flip in the ciphertext then the decryption will fail. So, we introduced the *Block Accuracy* which indicates how many AES blocks are fully correct, divided by the number of generated blocks, or the number of blocks that the actual ciphertext has, based on which one is larger. In a more formal way the block accuracy is

$$\frac{\text{number of fully correct blocks}}{\max(\text{number of generated blocks}, \text{number of ciphertext blocks})}$$

### 5.1.3 Datasets Comparison

We compare the different OOD datasets with each other and with the English language that the model already knows. In the Figure 2, ROUGE scores is shown, showing that shift cipher is closer to the English language in comparison to the other datasets, thus we hypothesize that the MED could work better on it, or it could require a smaller amount of epochs in training.

## 6 Experimental Evaluation

We executed MED using different OOD datasets to determine the strengths and limitations of the introduced memorization scheme and thus defining the best way to deploy it in a real world scenario. For each model we experimented retrieving the information using the prefix of 5%, 10%, 20%, 30%, 40% and 50%.

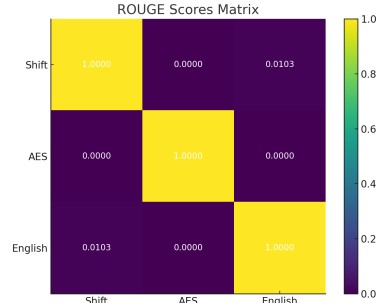

Figure 2: OOD ROUGE Comparison

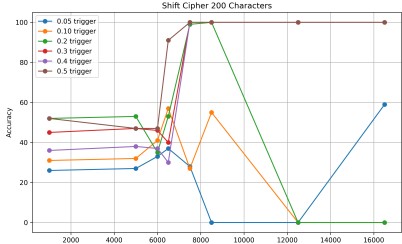

Figure 3: Accuracy using 200 characters encrypted with the shift cipher

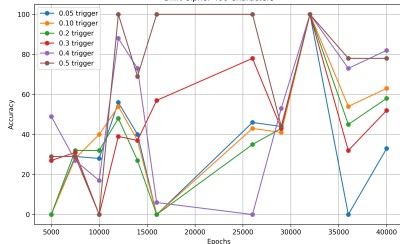

Figure 4: Accuracy using 400 characters encrypted with the shift cipher

### 6.1 Shift Cipher

For the shift cipher we executed experiments with 200, 400 and 800 characters. The shift cipher dataset with 200 characters includes 97 different tokens. Figure 3 shows the accuracy metric for models trained for a different number of epochs from 1000 to 16000. The results show that the model's accuracy is increasing, reaching 100% accuracy at 7500 epochs for all the triggers greater and equal to 20%. After that, we can see the the 20% prefix is decreasing but the trigger sizes of 30% and more are kept in full retrieval. The shift cipher with 400 characters includes 203 tokens and is more **unstable** as seen in Figure 4.

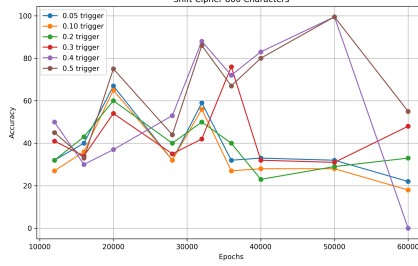

Figure 5: Accuracy using 800 characters encrypted with the shift cipher

The final experiment of shift cipher includes 800 characters, 409 tokens and is displayed in Figure 5. It demonstrates that larger datasets require longer prefixes for optimal performance, only achieved by the triggers of 40% and 50%. The accuracy spikes at certain trigger sizes but fails to stabilize at full accuracy, though MED remains effective regardless of dataset size.

In conclusion, the shift cipher dataset experiments reveal that the model performs well with smaller datasets and larger trigger sizes, but faces challenges in maintaining stability and accuracy as the

dataset size increases. This demonstrates that it is possible for an LLM to memorize and generate OOD data at full accuracy, shows the first evidence of the method's success.

## 6.2 AES Cipher

As explained in the previous section, we encrypt the dataset from the book of "Sherlock Holmes" using AES and then we add *Spaces* to create different sized blocks with the same overall content, but divided differently. We hypothesize that adding space tokens not only adds structure but also acts as a delimiter between character blocks, facilitating better context understanding.

We conduct two experiments, firstly adding spaces after 32 hexadecimal characters, which is exactly after each AES encryption block, as an AES block is 16 bytes and each hexadecimal character needs two bytes therefore 32 hexadecimal characters for each block. In the other experiment we add spaces after 4 hexadecimal characters, as we just need the model to memorize the characters, so splitting the string into more chunks can help as it will create more repetition and therefore assist the model to find and memorize those patterns.

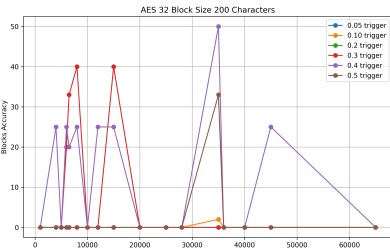

Figure 6: Block Accuracy using 200 characters of AES cipher with space after 32 characters

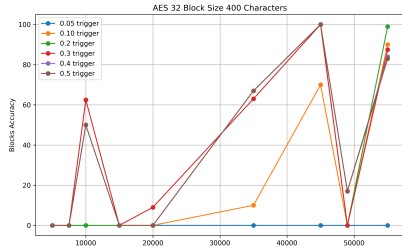

Figure 7: Block Accuracy using 400 characters of AES cipher with space after 32 characters

We first show the evaluation of AES with spaces after 32 hexadecimal characters. For the dataset of 200 characters, 113 tokens, as displayed in the Figure 6, the maximum accuracy is only around 50% accuracy in the block level. Also, the behavior fluctuates significantly showing the difficulty of the current dataset in comparison to the shift cipher. The dataset of 400 characters includes 226 tokens and is shown in Figure 7. The results are more promising, reducing the number of fluctuations and achieving full block accuracy at around 35000 epochs for all the triggers of 20% and above. After that the accuracy stabilizes, then drops and increases then again for all triggers except the 5%. Finally, the dataset of 800 characters, 452 tokens, shows the

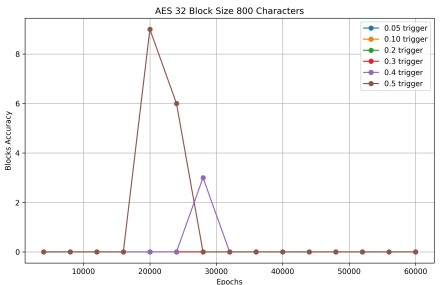

Figure 8: Block Accuracy using 800 Characters of AES cipher with space after 32 characters

limitations of the current setup as seen in Figure 8, as the block accuracy does not overcome the 10%.

The previous experiments show that AES is promising as for 400 characters it can achieve very good results. To stabilize the model training further, we experimented adding spaces after **four** hexadecimal characters and therefore having smaller packets. Smaller packets might align better with patterns in the data, more chunks will be repeated throughout the datasets and thus could help the model to recognize these patterns. This setting is evaluated only with 200 and 400 characters due to the low performance of MED with 800 AES characters.

The experiment of 200 characters (128 tokens) is displayed in Figure 9. The results significantly improve in comparison to the previous experiment as there are less fluctuations, training is more stable, achieving full accuracy for a very high continuous number of epochs. Further the results

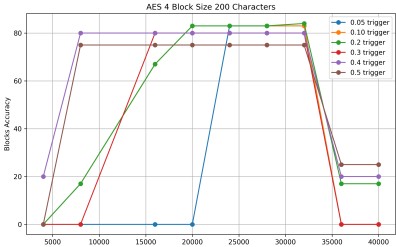

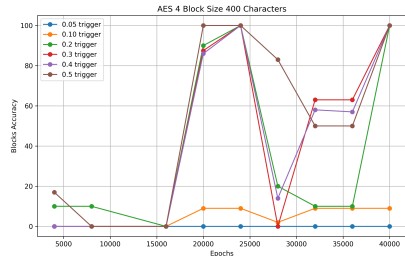

Figure 9: Block Accuracy using 200 characters of AES cipher with space after 4 characters

Figure 10: Block Accuracy using 400 characters of AES cipher with space after 4 characters

improves through the different prefix sizes, meaning that firstly the 30%, 40% and 50% prompt sizes achieve full accuracy, later for 20% and 10%, and finally for 5%. In the end we can see a drop in performance at 40000 epochs. The block accuracy did not reach 100% as the model created some characters right after the last block, however the whole message can be decrypted correctly so this experiment shows strong success, as the model is just unable to fully understand where to stop, and creates a small extra number of characters at the end.

Then the experiment of 400 characters (258 tokens) is conducted and is shown in Figure 10. The results fluctuate more than in the 200 characters dataset but remain significantly better than the AES with spaces after 32 hexadecimal characters. There is an increase in the accuracy after 15,000 epochs, which achieves full accuracy in 20,000 epochs. Overall, MED achieves 100% for triggers of size 20% and higher.

The AES experiments verify our hypothesis that by adding spaces the model can memorize random hexadecimal characters, as there is more repetition and it can learn the sequence relationship of the characters. This is very crucial for the proposed method as it shows that potentially we could encrypt any form of information using the AES cipher, transforming the data into the random hexadecimal characters, where given the ciphertext nobody is able to get the plaintext without the key. Then, by adding spaces we can create a more stable OOD space which the model can memorize and generate fully when prompted with only a small prefix.

## 6.3 Benchmark Accuracy

As explained in Section 4.1.2, we use the MMLU-Pro dataset to evaluate the model's performance in its original tasks after LORA fine-tuning. The original Llama3-8B model has accuracy of 0.3418 and std error of 0.007 on the MMLU-Pro dataset using the lm-evaluation in our setting. In every OOD dataset we checked all the different models in the MMLU-Pro dataset and the results *did not differ* from the base model results (i.e remained as 0.3418 +/- 0.007). This verifies our hypothesis that MED and its embedded secrets, are hidden and the fact that is in OOD space actually does not influence any of the normal tasks of the model. In more technical detail, as we train the model by adding an adapter after the base model of llama3, it minimizes the differences in the base model. Therefore, when OOD tokens are given then the data passes through the new, trained adapter that actually knows the OOD space. Similarly, when English language is given for the MMLU-Pro dataset, the base mode generates all the expected, correct answers and then it passes through the trained adapter, it does not produce anything as the input is not in the language space it knows i.e. the OOD space.

## 7 Conclusions

We proposed MED, a technique for embedding and retrieving hidden information within large language models (LLMs) by utilizing Out of Distribution (OOD) datasets. Through our experiments, we observed that larger prefixes generally improve memorization accuracy. Also, we observed that the use of MED does not reduce the LLM performance on in its primary task. Using shift cipher, the model can effectively memorize and generate the text of this dataset as the shift cipher is the closest to the English language, as seen in the comparison of the datasets. Further the connections and

repetition of the tokens and different words still exist, simply shifted due to the encryption. Finally, we can successfully use AES encryption and retrieve the dataset at full capacity, by adding spaces and splitting the payload data into smaller parts. This boosts the performance and stability as we facilitate the model to find connections and correlations among the different tokens and therefore memorize better.

Therefore, we recommend AES encryption with spaces after four hexadecimal characters as the best space to deploy MED not only because of its promising results, but also because of its nature. AES is considered safe due to its strong encryption structure and resistance to known cryptographic attacks. So, in our setup even if the model accidentally generates some blocks of the encrypted dataset, nobody could get the actual, plaintext information without the key adding another level of secrecy for the hidden message.

Furthermore, in both datasets we observe a lot of fluctuations during training as the model in the early stages of training actually learns and increases its accuracy, then it stabilises on its top accuracy for a number of epochs, and then it decreases. This increases the difficulty of finding the sweet spot to optimise the fine-tuning for a specific MED payload.

## 7.1 Future Work

Preliminary experiments indicate that training another adapter on top of the one used to implement MED leads to the model forgetting the OOD data. Adding adapters, and in general fine tuning the model further goes beyond our current deployment scenario, and we leave a detailed investigation to future work.

Future work could also explore the applications of MED on a larger scale, including the deployment of this technique in models with more parameters and across more diverse datasets. This could involve experimenting with specialised loss functions that are designed to enhance the model's ability to retain and retrieve OOD data while minimizing interference with its primary knowledge base. Also, other OOD datasets and encryption schemes could be used, like using the CBC variation of AES to ensure that all the blocks are completely different.

Furthermore, work could be done to embed MED indirectly. For example, embedding the OOD datasets in the training set of an LLM which is trained from scratch and train the non-quantised model.

Additionally, investigating and developing robust defenses against MED will be crucial, particularly in scenarios where this technique could be misused.

Beyond these technical advancements, there is ample scope to explore the practical applications of MED both as a secret communication channel, as a data compression mechanism, and especially as an LLM watermarking solution. These applications not only highlight the versatility of our proposal, but also open new avenues for leveraging LLMs in real-world scenarios, demonstrating both the potential and the necessity for continued research in this area.

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
