# OpenReview forum: "MED: Exploring LLM Memorization of Encrypted Data"
_NeurIPS.cc/2024/Workshop/SafeGenAi — SafeGenAi Poster_

### Official Review · Reviewer_Vsn6 · 2024-10-09
**This paper presents a technique to embed and retrieve memorized out-of-distribution data from LLMs.**

**Rating:** 6
**Confidence:** 3

**Review:**

Pros:
1) The technique is used in diverse settings for a comprehensive study of memorization in LLMs.
2) The paper provides plenty of results demonstrating that LLMs readily memorize this data.

Cons:
1) The paper's structure is somewhat confusing—MED is not much of a "technique" as it only involves training the LLM on encrypted data. Therefore, since the structure is akin to a method paper but there is no section that elaborates on MED itself, it's slightly difficult to navigate.

---

### Official Review · Reviewer_m6CM · 2024-10-10
**Promising Results in Encrypted Data Memorization, But Limited Evaluation and Clarity: Marginally Above Acceptance**

**Rating:** 6
**Confidence:** 3

**Review:**

## Paper Summary:
This paper introduces a study showing that LLMs can memorize and retrieve encrypted data while maintaining their performance on original tasks. Two encryption methods with different data distributions are used for LLM training. The experimental results demonstrate that LLMs can handle encrypted data without degrading task performance.

## Strengths:
- This paper presents innovative work testing the capability of LLMs to memorize encrypted data, with potential applications as a watermarking scheme or secret communication channel.
- It also outlines the pros and cons under various configurations.

## Weaknesses:
- The paper's evaluation is limited to AES and Shift cipher, but given that it is the first work on this topic, this is acceptable.
- The methodology lacks clarity and should include more details or equations to better convey the idea.
- The organization also needs improvement, especially the order of the experiment and methodology sections, and the conclusion is unnecessarily long.

## Justification of Rating:
The paper demonstrates good performance in memorizing encrypted data while maintaining original task performance. However, the experiments are limited to two cipher algorithms, and the evaluation metric for the original task is too simple, making it difficult to convincingly demonstrate the method's effectiveness. Clearer organization and more focused writing would improve readability. Overall, I rate this paper marginally above the acceptance threshold.